# GATA2 participates in protection against hypoxia-induced pulmonary vascular remodeling

Yuko Shirota[1�], Shin'ya Ohmori[2☉], James Douglas Engel[3], Takashi Moriguchi[🆔4☉*]

1 Division of Hematology and Rheumatology, Tohoku Medical and Pharmaceutical University, Sendai, Japan, 2 Department of Pharmacy, Faculty of Pharmacy, Takasaki University of Health and Welfare, Takasaki, Japan, 3 Department of Cell and Developmental Biology, University of Michigan, Ann Arbor, MI, United States of America, 4 Division of Medical Biochemistry, Tohoku Medical and Pharmaceutical University, Sendai, Japan

☉ These authors contributed equally to this work.
* moriguchi@tohoku-mpu.ac.jp

**Data Availability Statement:** Expression array data in HMVECs with siRNA against GATA2 (#GSE28304) and ChIP-seq data for GATA2 in HUVECs (#GSE29531), mast cells (# SRX206424),

## Abstract

The vascular endothelium is vital for cardio-pulmonary homeostasis and, thus, plays a crucial role in preventing life-threatening lung diseases. The transcription factor GATA2 is essential for hematopoiesis and maintaining vascular integrity. Heterozygous mutations in GATA2 can lead to a primary immunodeficiency syndrome with pulmonary manifestations. Some GATA2 haploinsufficient patients develop pulmonary hypertension (PH), characterized by vascular remodeling and occlusion of small pulmonary arteries. However, the mechanism underlying pulmonary vascular remodeling in GATA2 haploinsufficient patients remain unclear. To understand how GATA2 deficiency affects pulmonary artery homeostasis, we applied a chronic hypoxia-mediated PH model using inducible systemic *Gata2* conditionally deficient (*G2*-CKO) mice. The *G2*-CKO mice exhibited augmented pulmonary vascular remodeling, with enhanced α-smooth muscle actin accumulation and increased apoptotic cells in the vascular wall upon chronic hypoxia. Transcript analysis and chromatin immunoprecipitation assays using mouse pulmonary vascular endothelial cells revealed that GATA2 directly regulates the expression of *G6pdx* (a crucial cytoprotective enzyme) and *Bmp4* (a growth factor that mediates vascular homeostasis). These results suggest that GATA2-deficient lungs are vulnerable to the hypoxic stress due to a diminished cellular protective response, making *G2*-CKO mice more prone to vascular remodeling upon chronic hypoxia. These findings provide insights into the mechanisms underlying GATA2-haploinsufficiency-related pulmonary hypertension.

## Introduction

The pulmonary vasculature is essential for maintaining pulmonary circulation and gas exchange [1]. Impairment in this vital system disrupts cardiovascular homeostasis and can lead to life-threatening disorders. Pulmonary hypertension (PH) represents progressive

and ATAC-seq data from lug endothelial cells (# SRX11123445) were utilized. All the data are part of the ENCODE project (http://www.genome.gov/27528022). Publicly available data from the Human Protein Atlas can be found at https://www.proteinatlas.org/search/GATA2. Other data are available from the corresponding author upon request.

**Funding:** This work was supported by Grants-in-Aid for Scientific Research (22K06913 to TM) from the Ministry of Education, Culture, Sports, Science and Technology (MEXT) of Japan. The funders had no role in study design, data collection and analysis, decision to publish, or manuscript preparation.

**Competing interests:** The authors have declared that no competing interests exist.

vascular disease pathologically characterized by pulmonary vascular remodeling, specifically arterial wall thickening due to vascular fibrosis. These changes increase vascular resistance and pulmonary artery pressure, ultimately resulting in right ventricular failure [2]. The prognosis for untreated PH tends to be poor, even with current treatment regimens. The lung and heart status gradually worsens, accompanying sustained right heart failure and dyspnea, sometimes culminating in premature mortality [3, 4]. The etiological basis for PH has been the subject of numerous suggestive prior investigations. Genomic sequencing analysis has revealed that around 25–30% of idiopathic PH patients have underlying genetic causes [5]. Subsequent studies have identified several causal genes for these PH cases, which include loss of function mutations in the bone morphogenetic protein receptor 2 (BMPR2) locus and in related signaling molecules [6]. However, the mechanistic underpinnings for most idiopathic PH cases are yet to be discovered.

The GATA family of transcription factors each contains two C4 zinc fingers that serve as its DNA binding domain and recognizes the cognate consensus motif (A/T)GATA(A/G) [7, 8]. The zinc finger domain of the GATA factors are evolutionarily conserved among all six members (*Gata1~6*) that constitute this multigene family. GATA2, 3, and 6 expression is detected in vascular endothelial cells [9]. Among them, GATA2 is most abundantly expressed in endothelial cells throughout the vascular system [10]. GATA2 is known as an essential transcription factor for the development and maintenance of hematopoietic stem cells and multipotent progenitor cells [7, 8]. GATA2 also regulates endothelial-specific genes, including vascular endothelial cell adhesion molecule-1 (VCAM-1), endothelial nitric oxide synthase, van Willebrand factor, and endothelin in vascular endothelial cells [11]. Accumulating studies suggested that GATA2 might play a crucial role in developing and maintaining the vascular system. However, any pathophysiological roles for GATA2 in the adult cardio-pulmonary system have not been well characterized.

A decade of research has demonstrated that heterozygous mutations in GATA2 lead to a primary immune deficiency syndrome that affects hematopoiesis and the lymphatic and immune systems and is transmitted as an autosomal dominant trait or sporadic disease [12–14]. Pulmonary manifestations in the GATA2 deficiency syndrome are frequent and commonly involve nontuberculous mycobacterial infection. Notably, PH has been observed in 7–9% of GATA2 haploinsufficient patients, suggesting that GATA2-deficient patients are predisposed to this condition [14–17]. However, the precise mechanisms underlying PH in patients with GATA2-deficiency syndrome remain unknown. Given that GATA2 is abundantly expressed in vascular endothelial cells, we speculated that GATA2 loss might participate in the pathophysiology of pulmonary artery remodeling. To address this hypothesis, we subjected *Gata2* conditionally deficient mice (*G2*-CKO) to the chronic hypoxia-induced PH model. Our results demonstrate that the pulmonary artery of *G2*-CKO mice is vulnerable to hypoxic burden and, indeed, susceptible to extensive pulmonary artery remodeling, suggesting that maintaining an adequate level of GATA2 is a requisite for preventing PH.

## Results

### Expression profile of GATA2

GATA2 expression and function in the vascular endothelium have been mainly determined in cultured human endothelial cells [11, 18, 19]. However, GATA2 expression in the mammalian pulmonary vasculature has not been demonstrated. The public RNA-seq data of human lungs showed that lungs express a high level of GATA2 transcripts when compared to other tissues (Fig 1A) (www.proteinatlas.org/) [20]. Furthermore, single cell RNA-seq data of human lungs showed that GATA2 was highly expressed in pulmonary vascular endothelial cells among the

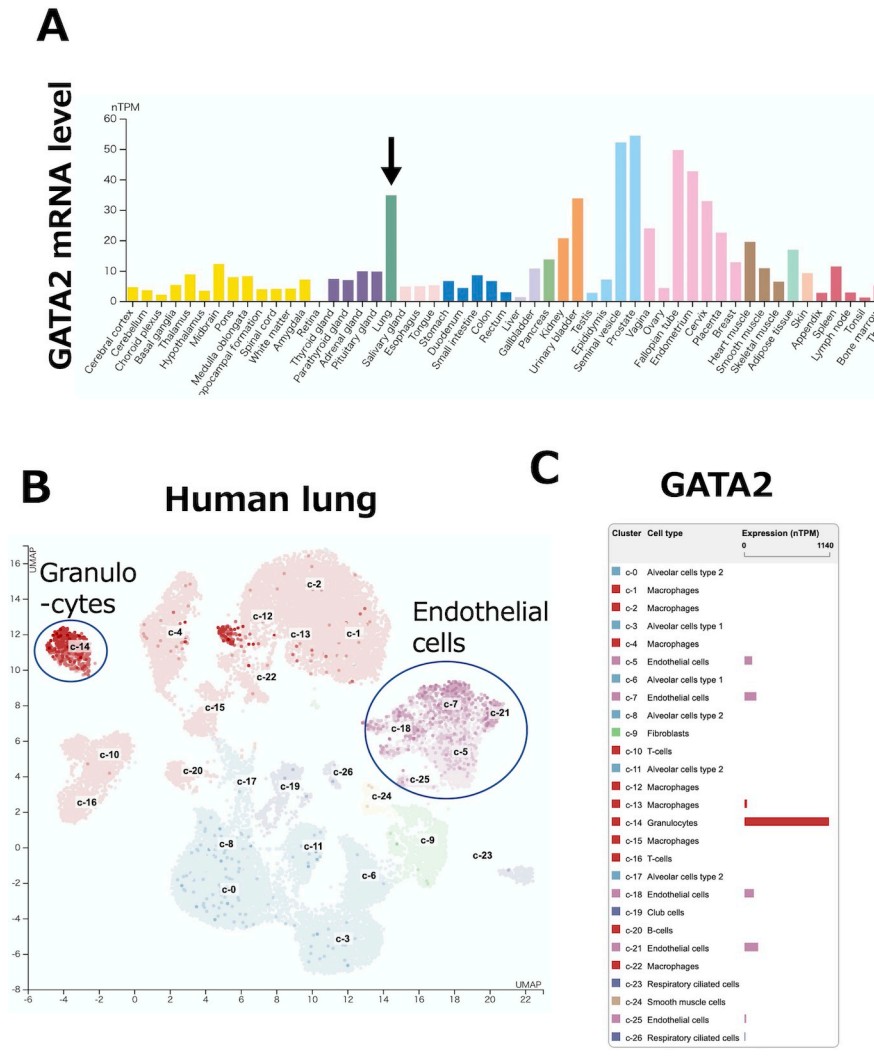

**Fig 1. Expression profile of GATA2. (A)** Human lungs (arrow) express relatively abundant GATA2 transcripts in comparison to many other tissues. **(B, C)** Single cell RNA-seq data for GATA2 expression in each cluster of cells from the human lung. Note that GATA2 is highly expressed in the pulmonary vascular endothelial cells (c-5, 7, 18, 21). Granulocytes (c-14) exhibit most abundant GATA2 expression. Reprinted from the Human Protein Atlas under a CC BY license (https://www.proteinatlas.org/ENSG00000179348-GATA2/single+cell+type/lung).

multiple cellular lineages (Fig 1B and 1C). Granulocytes, represented by mast cells, showed the highest GATA2 expression, consistent with the established function of GATA2 in mast cells [21, 22].

To clarify the histological distribution of GATA2-expressing cells in mammalian lung tissue, we employed *Gata2* GFP knock-in mice for histology. This strain of mouse harbors a GFP cassette inserted in the *Gata2* locus to trace the endogenous *Gata2* expression pattern [23]. We detected robust GFP-immunoreactivity in the endothelial cells lining the lumen of the pulmonary artery (Fig 2A and 2B). The granulocytes infiltrating into the alveolar interstitial spaces also showed intense GFP-immunoreactivity. When we double-stained the lung sections with anti-GFP- and anti-CD31/PECAM1(endothelial cell marker)-antibodies, the GFP signal overlapped that of CD31-positive vascular endothelial cells, confirming the endothelial cell-specific GATA2 expression (Fig 2C and 2D).

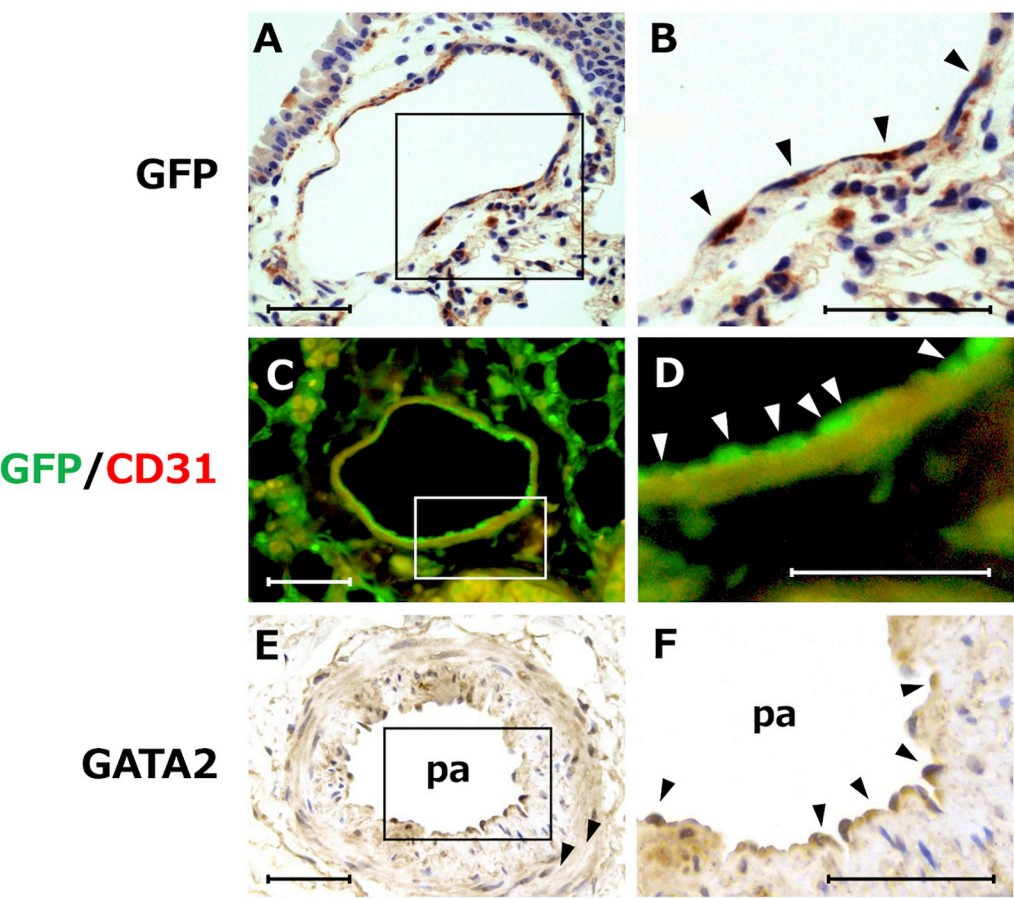

**Fig 2. GATA2 expression in lung vasculature. (A)** GFP immunoreactivity in the pulmonary arterial endothelial cells of *Gata2* GFP knock-in mice. **(B)** Higher magnification of the area within the rectangle in (A). GFP immunoreactivity is seen in the endothelial cells lining the lumen of the pulmonary artery (arrowheads). Granulocytes in the alveolar interstitial spaces also show GFP-immunoreactivity. **(C)** Coimmunofluorescence analysis of GFP and the CD31/PECAM1 endothelial marker on *Gata2* GFP knock-in mouse lung. Robust GFP expression was detected in the CD31-immunofluorescence-positive vascular endothelial cells. **(D)** Higher magnification of the rectangular area in (C). **(E)** GATA2 immunoreactivity in the pulmonary arterial endothelial cells from normal human lungs. **(F)** Higher magnification of the rectangular area in (E). Note that GATA2 immunoreactivity resides in the vascular endothelial cells (arrowheads). pa, pulmonary artery. Scale bars = 100 μm.

To validate these observations, we next stained normal human lung tissue using a GATA2-antibody. Consistent with the findings in mouse lungs, we found that the endothelial cells in the arterial lumen exhibited robust GATA2 immunostaining (Fig 2E and 2F). These results indicated that GATA2 is abundantly expressed in mouse and human lung pulmonary endothelial cells, implying that it may contribute to important physiological functions for homeostasis in the pulmonary vascular system.

## Hypoxia-induced pulmonary remodeling

Given the specificity of GATA2 expression in the pulmonary vasculature, we queried whether *Gata2*-deficiency might affect the severity of hypoxia-induced pulmonary arterial hypertension. To this end, we employed *Gata2* flox/flox (*G2*^f/f^) mice bearing the Rosa26-CreERT allele (*G2*^f/f^:: CreERT) (10–12 weeks). Mice were treated with 4-hydroxytamoxifen (4-OHT) to induce systemic *Gata2* deletion (referred to as *G2*-CKO or Group1); *G2*^f/f^::CreERT mice

without 4-OHT treatment served as controls (Group3). $G2^{f/f}$ mice not bearing the CreERT allele ($G2^{f/f}$ mice with or without 4-OHT treatment) were also used as controls (Groups2 and 4, respectively). We then exposed adult mice of all four genotypes (13–15 weeks old) to hypobaric hypoxia for 4 weeks to induce pulmonary hypertension.

By inspecting normal hematoxylin and eosin (HE) sections of the lung, we noticed that Group1(*G2*-CKO) mice showed more frequent pulmonary artery remodeling than Group3 (control) mice (Fig 3A). Elastica-Masson (EM) staining showed marked fibrosis in the remodeled arterioles of Group1 mice. In contrast, Group3 mice exhibited only moderate fibrosis in the arterioles (Fig 3A). The highly remodeled arterioles in Group1 mice were surrounded by dense α-smooth muscle actin (αSMA)-positive layers, indicating intense fibrotic changes in the arterial walls of the *G2*-CKO mice (bottom row in Fig 3A).

To quantitively evaluate the muscularization of the pulmonary arterioles, we classified the remodeled arterioles into three categories, i.e., "full" muscularization or "partial" muscularization, or "normal" (Fig 3B; see methods). Using this classification, we counted the number of vessels in each category in EM-stained lung sections. We found that the majority (over 70%) of the pulmonary arterioles in Group1 mice exhibited visible arterial wall remodeling: 30% of the arterioles showed full muscularization, and 40% showed partial muscularization of the arterial walls (Fig 3B). In contrast, the control groups (Groups 2, 3, and 4) showed only modest remodeling, with 10~15% of their arterioles showing full muscularization and 35–36% showing partial muscularization (Fig 3B). These results indicate that *G2*-CKO mice are more susceptible to arterial wall remodeling than are other group of control mice upon chronic normobaric hypoxia. Under normoxic conditions, all groups of mice exhibited normal vasculature to a similar extent: over 80% of their pulmonary arterioles were classified as normal, while the remaining small number of arterioles were partially remodeled (12–16%) or fully remodeled (2–3%) (S1 Fig). This control experiments confirms the successful induction of pulmonary arterial remodeling upon hypoxic exposure.

Hypoxia-induced PH is often associated with right ventricular hypertrophy (RVH), reflecting disease severity [24, 25]. Given this knowledge, we assessed RVH by examining the ratio of the right ventricle to the left ventricle plus the septum weight RV/(LV+S). However, we did not find significant exaggeration of the RVH in the *G2*-CKO mice (S2 Fig).

## Enhanced apoptosis in the *G2*-CKO mice

We next asked whether endothelial cell damage contributed to the enhanced pulmonary arterial remodeling in the *G2*-CKO mice. To this end, we subjected the lung sections to anti-cleaved caspase 3 immunostaining. As anticipated, the lung sections from Group1 mice showed more robust cleaved caspase 3-positive cells in the arterial intima than did Group3 control mice (Fig 4A). To further examine apoptotic status in these tissues, we conducted TUNEL staining. We found that more abundant TUNEL-positive cells were observed around the fully remodeled arterioles of Group1 mice than those in Group3 control mice, suggesting the apoptotic phenotype of GATA2-deficient vascular cells (Fig 4B).

## Inflammatory infiltration

Pulmonary arteriole damage is often associated with inflammatory cell infiltration [25]. Therefore, we assessed the infiltrating inflammatory cell population around the remodeled pulmonary artery. Immunohistochemical analysis of the lungs revealed an increased number of CD68+ inflammatory cells in the perivascular region of Group1 mice by comparison to Group3 control mice (Fig 5A). We enumerated the infiltrating CD68+ inflammatory cells surrounding arterioles in more than five consecutive slides across the lung and categorized them

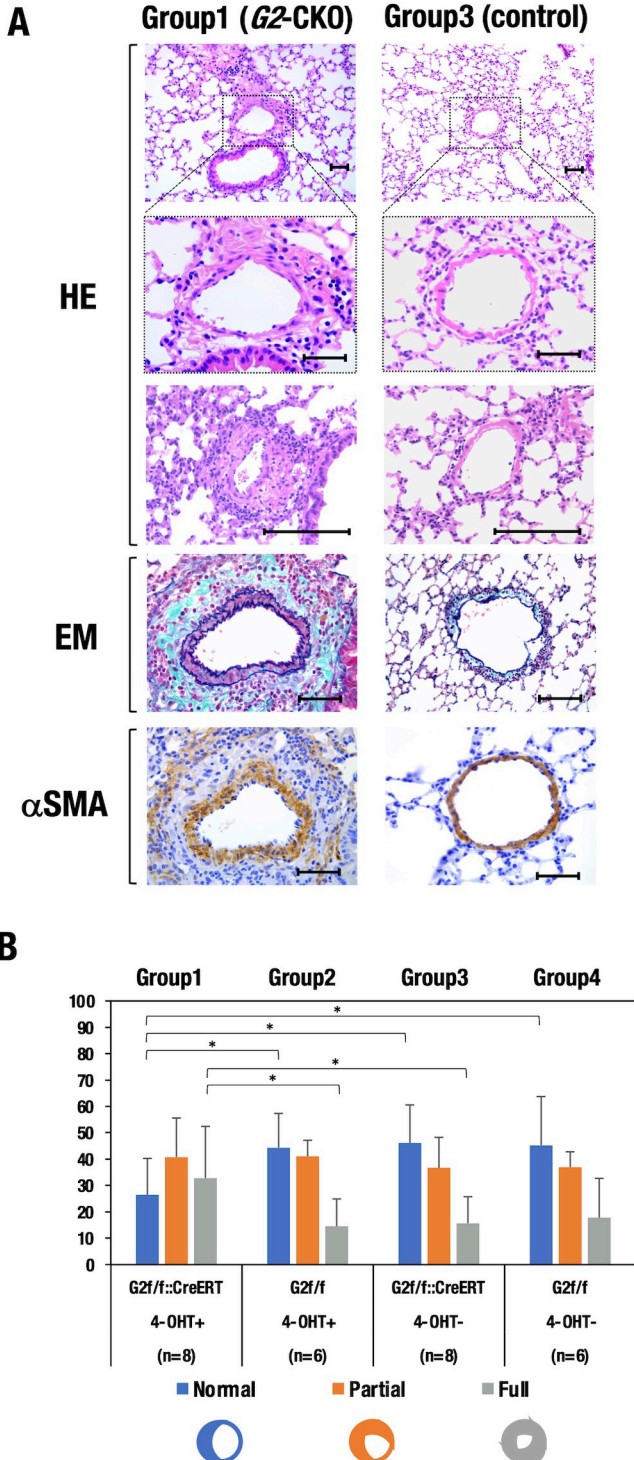

**Fig 3. Hypoxia-induced pulmonary remodeling. (A)** Group1 (*G2*-CKO) mice showed robust pulmonary artery remodeling upon chronic hypoxia compared to Group3 control mice ($G2^{f/f}$::CreERT without 4-OHT). HE, hematoxylin and eosin; EM, Elastica-Masson; αSMA, smooth muscle actin. Scale bars, 50 μm. **(B)** Quantitative evaluation of the pulmonary artery remodeling. Group1 mice showed a significant decrease in normal but increase in fully remodeled arterial walls. In contrast, Groups2, 3, and 4 control mice showed modest remodeling. Normal, arterioles with a double elastic lamina less than 50% of the arterial circumference; Partial, arterioles with distinct double elastic lamina visible in 50~80% of the circumference; Full, arterioles with distinct double elastic lamina visible in more than 80% of the arterial circumference. The genotype and number of each group of mice are depicted beneath the graph. Statistical significance between groups was assessed by Student's *t*-test. *; $P<0.05$). Scale bar = 50 μm.

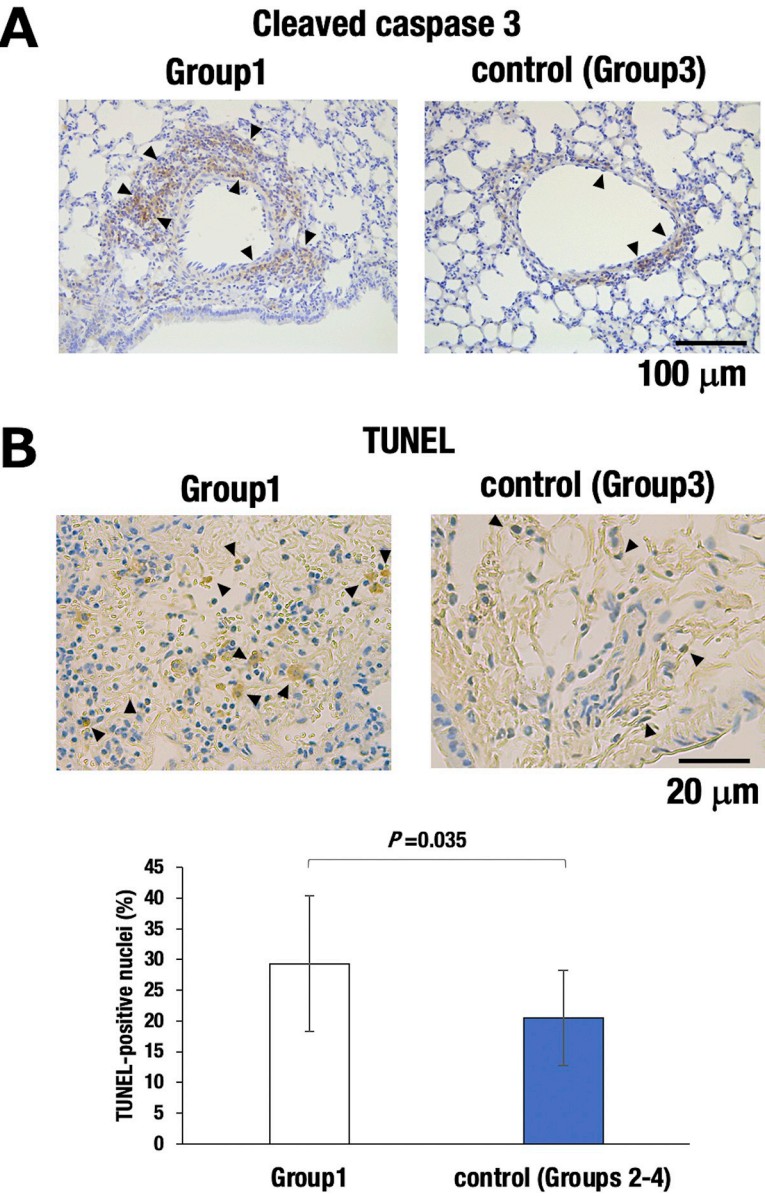

**Fig 4. Enhanced apoptosis in the *G2*-CKO mice. (A, B)** Representative microphotographs of cleaved caspase-8 and TUNEL stained lung sections from Group1 (*G2*-CKO) and Group3 control mice. Arrowheads indicate cleaved caspase-8- or TUNEL-positive cells. The lower diagram shows the percentage of TUNEL-positive cells around the arterioles in Group1 and control groups (mixture of Groups 2–4). A statistically significant difference is indicated. *; $P < 0.05$ vs. Group3 (two-tailed *t*-test).

into three states: minor, moderate, and severe, based on the degree of infiltration. After categorization, we calculated the ratio of the moderate and severe arterioles to all the vessels for each mouse lung. This ratio serves as a quantitative measure of the overall severity of inflammatory cell infiltration in lung tissue [25]. Consistent with the histochemical analysis, Group1 mice showed the most abundant infiltrating inflammatory cells compared to the other control mice (Fig 5B). These results suggest that the vascular endothelial cells of *G2*-CKO mice are prone to hypoxia-induced apoptotic damage and subsequent inflammatory insults.

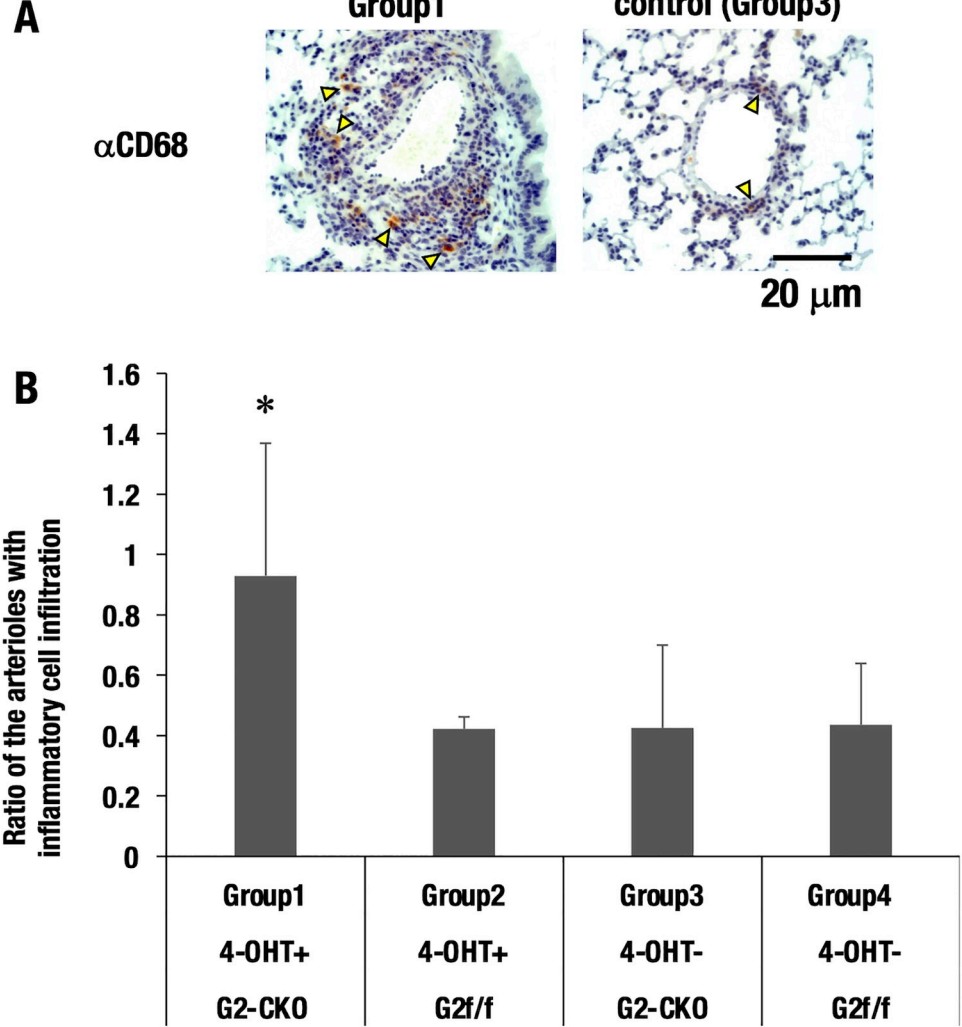

**Fig 5. Inflammatory infiltration. (A)** Group1 (*G2*-CKO) mice showed a greater number of CD68+ inflammatory cells in the perivascular region than Group3 control mice (see arrowheads). CD68 predominantly labels monocytes and macrophages. (B) Group1 mice showed the most abundant inflammatory cell infiltration compared with the other group of control mice. *; $P<0.05$ vs. Group3 (two-tailed *t*-test).

### Gene expression profile in the HMVEC cells

To delve into the mechanism underlying the enhanced vulnerability of *G2*-CKO mice to hypoxia-induced pulmonary artery remodeling, we exploited a microarray dataset of human microvascular endothelial cells (HMVECs) treated with siGATA2 RNA (accession#; GSE28304) [11]. A list of 788 (upregulated) and 818 (downregulated) genes that were significantly affected by siGATA2 knockdown was subjected to functional clustering using the Database for Annotation, Visualization, and Integrated Discovery (DAVID; http://david. abcc. ncifcrf.gov/) and Metascape (https://metascape.org/gp/index.html#/main/step1). This combined pathway analysis categorized the 1608 differentially expressed genes (DEGs) into several gene ontology (GO) terms, including vascular development, cell migration, programmed cell death, apoptosis, proliferation, and response to stimulus (Fig 6A). When the network of the GO terms was visualized via Cytoscape (https://cytoscape.org/), the enriched pathways were highly clustered (Fig 6B and 6C).

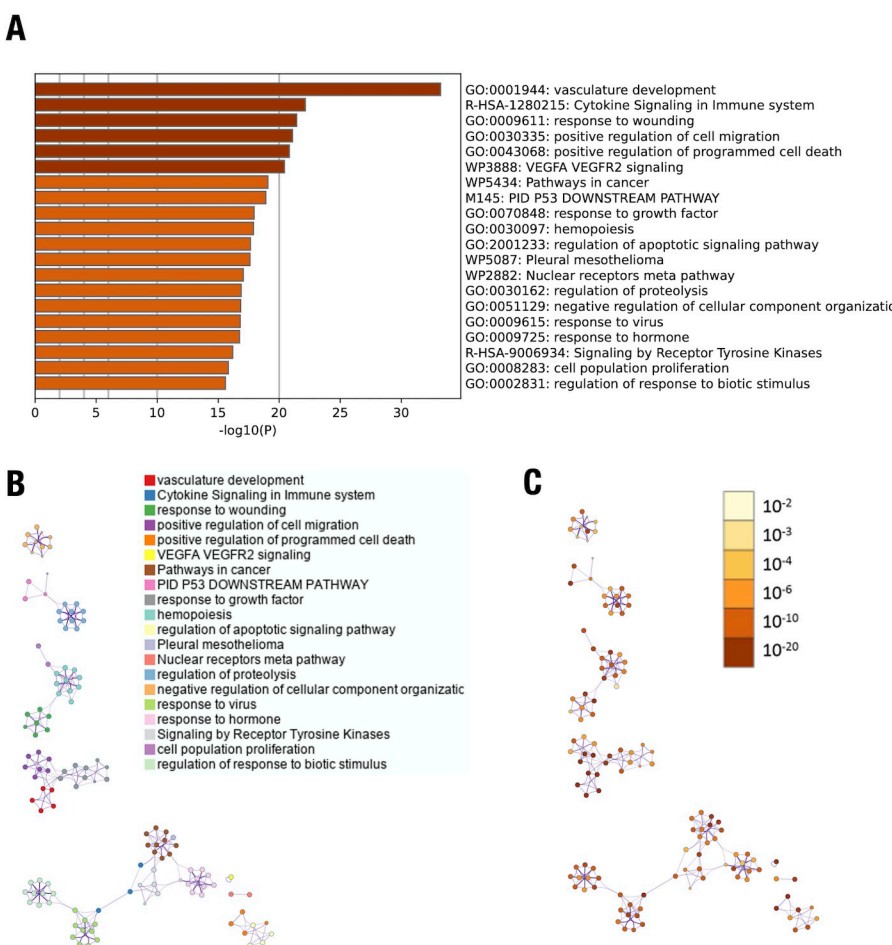

**Fig 6. Gene expression profile in the HMVEC cells.** (**A**) Bar graph of enriched terms of the differentially expressed genes (DEGs) upon siGATA2 in HMVEC cells (colored by *p*-values). Gene ontology terms related to vascular development, cell migration, programmed cell death, apoptosis, proliferation, and response to stimulus are enriched by siGATA2 in HMVEC cells (highlighted in yellow). (**B**) Network of the enriched terms (colored by cluster ID) of the DEGs. (**C**) Network of the enriched terms (colored by *p*-value) of the DEGs. The charts were generated using expression array data from HMVECs with a siRNA against GATA2 (GSE28304) and processed through Metascape, a gene annotation public resource (http://metascape.org).

## Gene expression profile in vascular endothelial cells of the lung

Among the genes annotated in these pathways, we examined the mRNA expression levels of *G6pdx*, *Tgfb1*, *Tgfbr1*, *Hmga2*, *Bmp4*, and *Atf3*. All of these genes have been implicated in cardiovascular diseases and pulmonary fibrosis [5]. To more precisely refine which of these genes might participate in pulmonary vascular remodeling, we sorted out the CD31⁺/CD45⁻ non-hematopoietic vascular endothelial cells from the mouse whole lung single cell population by flow cytometry (Fig 7A). We found that mRNA expression of *G6pdx* and *Bmp4* was significantly lower in *G2*-CKO lung endothelial cells (Fig 7B). This decrease aligns with a similar trend observed following siGATA2 knockdown in HMVECs (Fig 7C). Expression of *Tgfb1*, *Tgfbr1*, *Hmga2*, and *Atf3* was also diminished in *G2*-CKO endothelial cells, regardless of statistical significance (Fig 7B). In contrast, mRNA expression of these genes increased in the siGATA2-treated HMVECs, possibly representing differences between cultured human endothelial cells and directly isolated mouse pulmonary endothelial cells (Fig 7C).

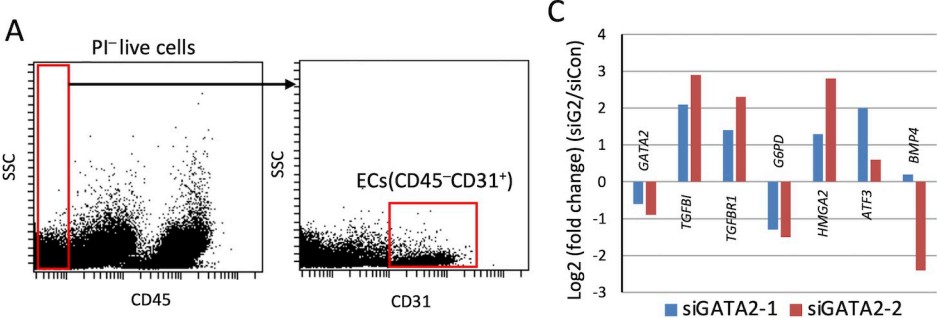

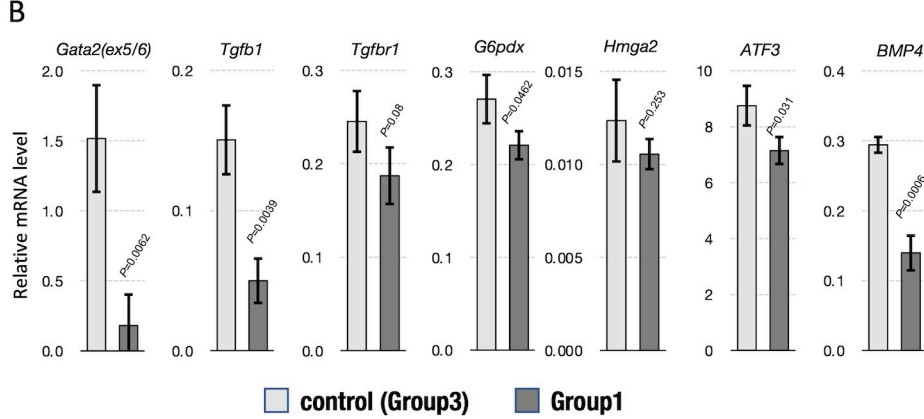

**Fig 7. Gene expression profile in vascular endothelial cells of the lung. (A)** Sorting strategy for CD31+/CD45⁻ pulmonary vascular endothelial cells from the mouse lung (Group1). **(B)** Gene expression profiles of CD31+/CD45⁻ pulmonary vascular endothelial cells from Group3 (control) and Group1 (*G2*-CKO) mice. Statistical significance between groups was assessed by Student's *t*-test. At least three mice from each group are analyzed. **(C)** Changes in the gene expression implicated with pulmonary hypertension in HMVECs. The Log2 ratio of mRNA levels under siGATA2 knockdown compared to those under control random siRNA was calculated to yield the fold change. Expression array data from HMVECs with two independent siRNA against GATA2 (#GSE28304) were utilized.

## GATA2 regulates *Bmp4* and *G6pdx*

Given the known cytoprotective activity of *Bmp4* and *G6pdx* in the pulmonary vasculature [26, 27], we examined the potential contribution of these two genes to GATA2-mediated pulmonary vascular homeostasis. To investigate clues regarding possible GATA2 function at these loci, we exploited GATA2 ChIP-seq data of mast cells (accession# SRX206424) and assay for transposase-accessible chromatin using sequencing (ATAC-seq) data of lung endothelial cells (accession# SRX11123445). Robust GATA2 binding was detected at intron and promoter sequences of these gene loci (Fig 8A). Notably, ATAC-seq peaks in mouse lung endothelial cells partially overlapped GATA2 binding in the *Bmp4* 1ˢᵗ intron and *G6pdx* promoter, suggesting that GATA2 directly regulates *Bmp4* and *G6pdx* through these presumptive regulatory elements (Fig 8A). To demonstrate direct GATA2 binding to these regions, we conducted chromatin immunoprecipitation experiments using the CD31⁺/CD45⁻ pulmonary vascular endothelial cells of wild-type mice. JASPAR (http://jaspar. genereg.net), an open-access database for transcription factor binding sequences, predicted several consensus GATA-binding sites in the *Bmp4* and *G6pdx* loci (Fig 8B). Consequently, we designed PCR primer pairs to amplify these sequences in immune precipitated

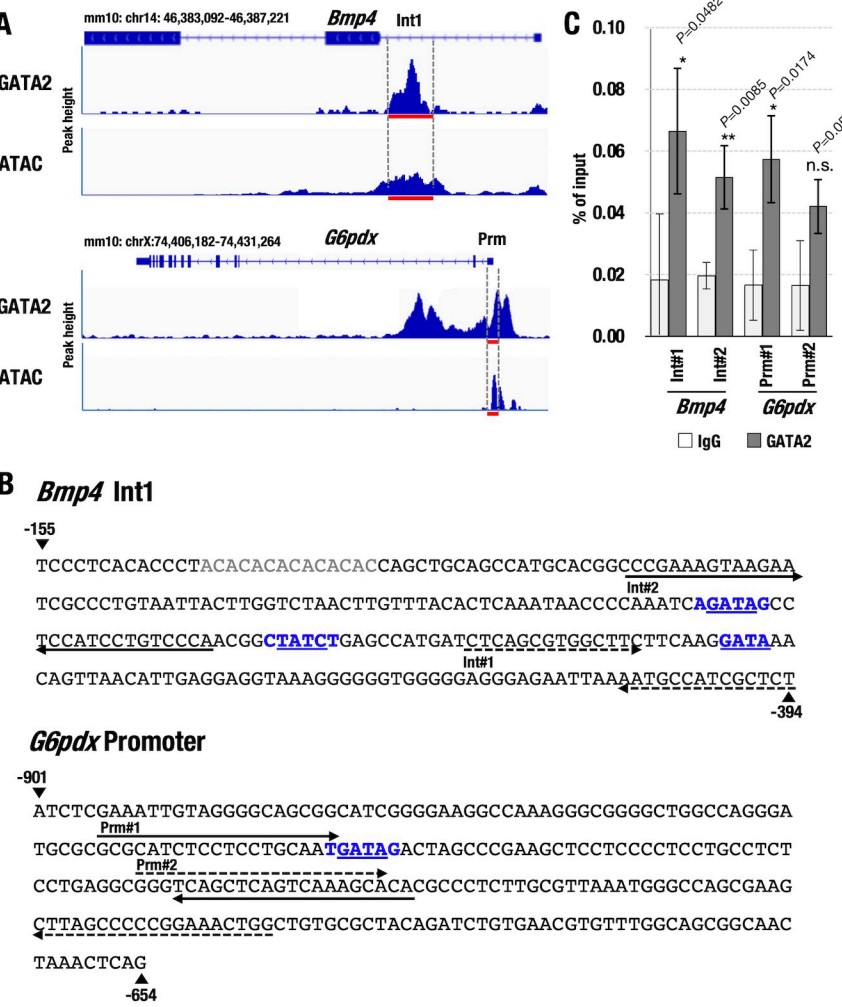

**Fig 8. GATA2 regulates *Bmp4* and *G6pdx*. (A)** Robust GATA2 binding peaks at the *Bmp4* and *G6pdx* loci in mouse mast cells. Note that ATAC-seq peaks in lung endothelial cells partially overlap with the GATA2 binding peaks at the *Bmp4* 1st intron and *G6pdx* promoter sequences. GATA2 ChIP-seq data from the mast cells (#SRX206424) and ATAC-seq data from the lung endothelial cells (#SRX11123445) are obtained from the NCBI Sequence Read Archive (SRA) database. **(B)** Nucleotide sequences of the GATA2 binding peak regions in the *Bmp4* 1st intron and *G6pdx* promoter regions. The primer positions and consensus GATA2 binding sites are indicated. **(C)** Quantitative chromatin immunoprecipitation (ChIP–qPCR) analysis of GATA2 at the *Bmp4* 1st intron and *G6pdx* promoter regions in CD31+/CD45− pulmonary vascular endothelial cells. Note that GATA2 robustly binds to the promoter and 1st intron regions depicted in **(A)**.

chromatin DNA. The results showed that GATA2 bound to the *Bmp4* and *G6pdx* loci in native pulmonary vascular endothelial cells (Fig 8C, suggesting that GATA2 directly regulates the expression of these two genes.

Public ChIP-seq data of human umbilical vein endothelial cells (HUVECs) also showed robust GATA2 binding in the vicinity of the *G6PD* and *BMP4* locus, supporting the contention that there is a direct regulatory influence of GATA2 on these genes in HUVECs (accession #SRX070876) (S3 Fig). These results suggest that GATA2 regulates *G6pdx* and *Bmp4* expression in pulmonary vascular endothelial cells and potentially contributes to the prevention of hypoxia-induced pulmonary vascular remodeling.

## Discussion

Numerous studies have demonstrated that GATA2 plays a crucial role in the maintenance and function of vascular endothelial cells. However, the specific roles played by GATA2 in the pulmonary vasculature remain to be elucidated. Given the high expression levels of GATA2 in the pulmonary endothelial cells, we hypothesized that GATA2 is pathophysiologically involved in the response observed in the murine pulmonary hypertension (PH) model. The results of studies presented here partially support this hypothesis. The GATA2-deficient Group1 mice showed more severe pulmonary artery remodeling upon hypoxic exposure. The accelerated pulmonary artery remodeling in Group1 mice was associated with increased αSMA accumulation and increased numbers of apoptotic cells in the vascular intima. These results demonstrate that GATA2 plays crucial protective roles against hypoxia-induced pulmonary artery remodeling in mice. The biological pathway analysis of the public transcriptome data in the GATA2--knockdown HMVECs highlights that GATA2-regulated genes are enriched in vascular development, cell migration, programmed cell death, apoptosis, proliferation, and response to stimulus. Indeed, the expression of several of these genes was altered in the GATA2-deficient lungs, suggesting that these GATA2-regulated endothelial genes were potentially involved in the pathogenic response to hypoxia-induced pulmonary remodeling. Despite these significant findings, the present study did not demonstrate PH susceptibility in the GATA2-deficient mice. A key limitation of our study is the absence of direct right ventricular systolic pressure (RVSP) measurements, which are crucial for diagnosing PH. Future studies should incorporate right heart catheterization to confirm elevated RVSP and provide stronger evidence for PH susceptibility in the GATA2-deficient mice.

BMP4, which is expressed in the adult mouse lung, and particularly in endothelial cells, is induced by hypoxia [28]. BMP4 protects rat pulmonary arterial smooth muscle cells from apoptosis and prevents pulmonary fibrosis [26, 29]. Genetic studies in patients with familial primary pulmonary hypertension (FPPH) have identified frequent mutations in BMPR2 [6]. Since BMP4 binds to a heterodimerized complex of BMPR1 and BMPR2 and transmits downstream signals, BMPR2 mutations might retard BMP4 signaling. Han *et al.* demonstrated that SMAD1 deficiency in either endothelial or smooth muscle cells can predispose mice to pulmonary hypertension due to an impaired BMP4 signaling pathway, suggesting that the BMP downstream mediator (i.e., SMAD1) is also crucial for the BMPR2 signal and the prevention of pulmonary artery hypertension [30]. Given these prior observations, we surmise that lower levels of BMP4 might predispose the GATA2-deficient Group1 mice to hypoxia-induced pulmonary arterial remodeling.

We have previously reported that GATA2 can stimulate BMP4 expression in the urogenital primordium [31]. Another group reported that GATA2 and GATA3 upregulate BMP4 expression in hemangioblasts and human embryonic stem cells [32] [33]. The GATA2-BMP4 axis might also operate in endothelial cells to maintain vascular homeostasis. In the present study, we demonstrate that GATA2 directly binds to multiple GATA binding sites in the *G6pdx* and *Bmp4* loci in mouse pulmonary endothelial cells. These results indicate that GATA2 could directly regulate *G6pdx* and *Bmp4* expression in mouse pulmonary vascular endothelial cells. GATA2 ChIP-seq data also showed binding peaks in the flanking region of the *BMP4* and *G6PD* loci in HUVECs, suggesting a direct regulatory influence of GATA2 over BMP4 in the human vascular endothelial cells.

Chronic hypoxic exposure induces cellular stress on endothelial cells, leading to pulmonary vascular remodeling, which recapitulates the etiology of human pulmonary hypertension [34]. Presumably, the perturbed defense system in the GATA2-deficient Group1 mice led to vulnerability due to increased hypoxic burden, which increased endothelial cell apoptosis and

accelerated arterial remodeling. Consistent with this hypothesis, we found that Glucose-6 phosphate dehydrogenase (G6PD), a rate-limiting enzyme in the pentose phosphate pathway, was decreased in the lungs of Group1 mice. G6PD plays a crucial role in the cellular oxidative response by supplying nicotinamide adenine dinucleotide phosphate (NADPH) to recycle glutathione. Indeed, several reports have shown that some patients with idiopathic pulmonary hypertension have various types of G6PD deficiency [27]. It has also been demonstrated that G6PD plays a protective role in vascular endothelial cells [35]. G6PD-deficient mice develop spontaneous pulmonary hypertension with pulmonary artery remodeling, which is associated with increased oxidative stress [27]. G6PD is essential for protecting erythrocytes from hemolysis by ameliorating oxidative stress. Indeed, G6PD deficiency represents one of the most common erythrocyte disorders, affecting approximately 5% of the global population. G6PD expression is under the regulatory influence of GATA1 in erythroid cells [36]. By analogy to this epistatic relationship, GATA2 might regulate G6PD expression and protect the endothelial cells from oxidative damage.

Human vascular endothelial cells express GATA2, -3, and -6 [9]. GATA2 is most abundantly expressed throughout in large to small vessels. Although endothelial GATA6 abundance is lower than GATA2 and GATA3, a previous study showed that GATA6 deficiency in the pulmonary vascular cells evokes oxidative stress, potentially contributing to PH development [37]. The same study showed that GATA6 acts as a direct positive regulator of antioxidant enzymes, i.e., glutathione peroxidase (GPX), superoxide dismutase (SOD) and catalase [37]. These results imply the potential involvement of GATA factors in oxidative stress response mechanisms in multiple cellular contexts.

Overall, these new data in concert with prior studies suggest that GATA2 plays a pivotal role in regulating pulmonary vascular homeostasis and that GATA2 deficiency promotes susceptibility to pulmonary vascular remodeling and subsequent PH by disturbing the cellular maintenance and defense systems. There are currently no effective therapeutics to prevent or treat PH. Therefore, intense efforts have been made to elucidate the etiology of PH to identify novel therapeutic targets. Only through such molecular and genetic studies will we adequately understand the pathogenic mechanisms leading to PH. Ideally, these efforts will lead to the identification of additional novel targets for new therapeutics.

## Materials and methods

### Mice

*Gata2*flox/flox mice (a kind gift from S. A. Camper, University of Michigan) backcrossed to a C57BL/6J congenic strains for over ten generations were bred with mice expressing tamoxifen-inducible Cre recombinase controlled by the Rosa 26 promoter (CreERT mice; The Jackson Laboratory) (Charles et al., 2006). CreERT transgene induction was performed as previously described. Briefly, *Gata2*f/f::CreERT mice (10–12 weeks old) received intraperitoneal injections of 1μg tamoxifen (Sigma-Aldrich) on Days 1– to 3 and 8– to 10, and were subsequently used on Days 20–22 after the first injection. Mice used in experiments were matched by weight (15.1–27.0 g) and age (13–15 weeks).

The *Gata2* GFP knock-in (*Gata2*GFP) allele includes a GFP reporter gene inserted at the *Gata2* translational initiation site, disrupting normal GATA2 translation. Heterozygous *Gata2*GFP mice survive to adulthood, while homozygous *Gata2*GFP embryos exhibit lethal hematopoietic defects [23]. All mice were maintained in a specific pathogen-free facility. Primers used for genotyping are listed in Table 1. All animal experiments were conducted in accordance with protocols approved by the Institutional Animal Care and Use Committee of Tohoku University and Tohoku Medical and Pharmaceutical University (Permit Number:

**Table 1. Sequences of primers used for genotyping, RT-qPCR, and ChIP-qPCR.**

| Gene | Sense primer | Antisense primer | Assay |
|---|---|---|---|
| Gata2<sup>flox</sup> | TCCGTGGGACCTGTTTCCTTAC | GCCTGCGTCCTCCAACACCTCTAA | genotyping |
| ROSA26CreER | CGGTCTGGCAGTAAAAACTAT | CAGGGTGTTATAAGCAATCCC | genotyping |
| Gata2<sup>GFP</sup> | CTGAAGTTCATCTGCACCACC | GAAGTTGTACTCCAGCTTGTGC | genotyping |
| Gata2 ex5/6 | GCACCTGTTGTGCAAATTGT | GCCCCTTTCTTGCTCTTCTT | RT-qPCR |
| G6pdx | TGTGGAGAATGAACGGTGGG | AAGATGTCACCTGCCACGTC | RT-qPCR |
| Tgfb1 | TCAAACAGGCGTCAGCGTAT | TCTCCTCCTCGGTCTTCCTG | RT-qPCR |
| Tgfbr1 | GCATTGGCAAAGGTCGGTTT | TGCCTCTCGGAACCATGAAC | RT-qPCR |
| Hmga2 | CCCTCTAAAGCAGCCCAGAA | TGCGAGGATGTCTCTTCAGT | RT-qPCR |
| Atf3 | GCTGAGATTCGCCATCCAGA | TGACATCTCCAGGGGTCTGT | RT-qPCR |
| Bmp4 | AGCCAACACTGTGAGGAGTTT | AAAGCAGAGCTCTCACTGGT | RT-qPCR |
| Polr2a | CTGGACCCTCAAGCCCATACAT | CGTGGCTCATAGGCTGGTGAT | RT-qPCR |
| Bmp4 Int#1 | TGAGCCATGATCTCAGCGTG | CGAAGTGAAGAGCGATGGCA | ChIP-qPCR |
| Bmp4 Int#2 | CCCGAAAGTAAGAATCGCCC | GATAGCCGTTGGGACAGGAT | ChIP-qPCR |
| G6pdx Prm#1 | CATCTCCTCCTGCAATGATA | GGGCGTGTGCTTTGACTGAG | ChIP-qPCR |
| G6pdx Prm#2 | CAGCTCAGTCAAAGCACACG | CACAGCCAGTTTCCGGGG | ChIP-qPCR |

19017-cn) and complied with the Animal Research: Reporting of In Vivo Experiments (ARRIVE) guidelines.

## Hypoxic exposure

GATA2-deficient and control mouse groups (12- to 15-week old) were exposed to either hypoxia or normoxia for 4 weeks as previously described. Hypoxic mice were housed in an acrylic chamber with a nonrecirculating gas mixture of 10% O2 and 90% N2, provided by adsorption-type oxygen concentrator (Teijin, Japan). Normoxic mice were housed in room air (21% O2) under a 12-hour light-dark cycle. Hypoxic exposure was achieved using a hypobaric chamber (Shizume Medical, Tokyo, Japan) set to simulate an altitude of 17,000 ft [barometric pressure (Pb) = 410 mmHg; inspired PO2 = 76 mmHg], as previously described [25]. All mice were maintained in a specific pathogen-free facility. Experimental protocols were approved by the Animal Care Committee of Tohoku University and Tohoku Medical and Pharmaceutical University (Permit Number: 19017-cn)

## Histological analyses of pulmonary arteries

Under isoflurane anesthesia (1.0%), mice were perfused with phosphate-buffered saline (PBS) and subsequently fixed with 10% phosphate-buffered formalin for 5 min. The lungs were dissected, fixed for 24 hours, and embedded in paraffin. Lung cross-sections (4 μm thick) through the hilum were subjected to standard Elastica–Masson (EM) staining. Pulmonary arteries adjacent to airways distal to the respiratory bronchiole were examined microscopically for arterial remodeling based on the circumferential EM staining pattern (blue), with modifications as reported previously [25]. Briefly, if they showed.

Arterioles with a distinct double elastic lamina visible in more than 80% of the circumference were designated "fully muscularized". Those with a double elastic lamina visible in 50–80% of the circumference were classified as "partially muscularized", while arteries with a double elastic lamina covering less than 50% of the circumference were considered normal. The percentage of vessels in each category was calculated relative to the total number of vessels. For each mouse, a total of 60–80 vessels from at least 20 different fields were examined under a microscopes (Leica DM2500, Tokyo, Japan).

For immunohistochemistry, paraffin sections (4 μm) underwent standard antibody staining with the following primary antibodies: rabbit anti-GATA 2 (11103-1-AP; ProteinTech), chicken anti-GFP (ab13970; Abcam), rabbit anti-CD68 (ab125212; Abcam), rabbit anti-CD31 (ab28364; Abcam), and anti-Cleaved Caspase-3 (ab2302; Abcam). Formalin-fixed paraffin-embedded normal human lung tissues were obtained from Proteogenex Inc. (Culver City, CA). The Ethics Committees at Tohoku Medical and Pharmaceutical University (Sendai, Japan) approved the research protocols for this study (2021-4-066).

## Assessment of right ventricular hypertrophy

Right ventricular hypertrophy was assessed as described previously [25]. Briefly, the mice were exsanguinated, and the hearts were isolated and fixed with 10% phosphate-buffered formalin. Formaldehyde-fixed dry hearts were dissected, and the right ventricular wall was removed from the left ventricle and septum. The ratio of the right ventricle to the left ventricle plus septum weight [RV/(LV+S)] was calculated to determine the extent of right ventricular hypertrophy.

## Apoptosis of pulmonary arterial cells

To estimate the degree of apoptosis in pulmonary arterial cells, we performed a terminal deoxynucleotidyl-transferase-mediated dUTP nick end-labeling (TUNEL) assay on paraffin sections (4 μm) according to the manufacturer's instructions (Promega, Madison, WI, USA). In TUNEL-stained lung sections from 3 to 4 mice per group, we randomly selected at least ten fields from each group and counted the number of nuclei in the vascular wall. The results are expressed as a percentage of the number of TUNEL-positive nuclei relative to the total number of nuclei. Statistical significance between groups was assessed using Student's *t*-test.

## RT–PCR analysis

The mRNA expressions of vascular homeostasis-related genes was evaluated via quantitative RT–PCR (RT–qPCR). The G2-CKO vascular endothelial cell (EC) isolation was performed following the manufacturer's instructions for the Lung Dissociation Kit (Miltenyi Biotech, 130-095-927). In brief, the lung tissues were first mechanically dissociated and enzymatically digested using gentleMACS Octo Dissociator (Miltenyi Biotech). For further isolation of ECs, lung cells were stained for CD45 and CD31. CD45(-)/CD31(+) ECs were sorted on a FACSJazz cell sorter (BD Biosciences). Total RNA was prepared with NucleoSpin RNA Plus (TaKaRa) according to the manufacturer's instructions. The RNA samples were reverse transcribed using a ReverTra Ace qPCR RT kit (Toyobo) following the manufacturer's instructions. For RT–qPCR, cDNA was analyzed using a Thermal Cycler Dice Real-Time System III (TAKARA) with THUNDERBIRD Next SYBR qPCR Mix (TOYOBO) according to the manufacturer's instructions. The data were normalized to *Polr2a* expression levels and are shown as the means ± the standard deviations (SDs). Statistical significance between groups was assessed by Student's *t*-test.

The primers used for genotyping and qRT–PCR are listed in Table 1.

## Chromatin immunoprecipitation (ChIP) assay

The ChIP assay and quantitative analysis of DNA purified from the ChIP samples were conducted as described previously (Ohmori et al., 2012; Ohmori et al., 2021). CD45(-)/CD31(+) lung endothelial cells from the wild-type mice were sorted as described above in the method for RT–PCR analysis. The cells were crosslinked with 1% formaldehyde for 10 min and lysed.

The chromatin samples were sonicated to shear the DNA using a focused ultrasonicator M220 (Covaris) with a duty factor of 5%, a peak incident power of 75 W, and 200 cycles per burst for 300 sec. The solubilized chromatin fraction was incubated with the primary antibodies overnight, which were prebound to anti-mouse IgG-conjugated Dynabeads (Thermo Fisher Scientific). The primary antibody used for the ChIP assay was anti-GATA2 (B9922A; Perseus Proteomics). The DNA purified from the ChIP samples was decrosslinked, purified, and subjected to analysis using a Thermal Cycler Dice Real-Time System III (TAKARA) with THUNDERBIRD Next SYBR qPCR Mix (TOYOBO). The primers used for the ChIP assays are listed in the Table 1.

### Statistical analyses

All values are expressed as means ± standard deviations (SD). Statistical significance between groups was assessed by two-tailed Student's $t$-test, unless otherwise specified. A $P$-value $< 0.05$ was considered statistically significant.

### Supporting information

**S1 Fig. Pulmonary artery vasculature under normoxic condition.** Under normoxic conditions, all group of mice, including Group1 (*G2*-CKO), exhibited normal vasculature to a similar extent: more than 80% of their pulmonary arterioles were normal, while the remaining arterioles were partially remodeled (12–16%) or fully remodeled in a small number of cases (2–3%).
(TIFF)

**S2 Fig. The ratio of the right ventricle to the left ventricle plus septum weight [RV/(LV +sep)].** Group1 mice (*G2*-CKO) show comparable levels of right ventricular wall thickness compared with other group of control mice.
(TIFF)

**S3 Fig. GATA2 binding peaks in the vicinity of the *G6PD* and *BMP4* loci in HUVECs.** The publicly available ChIP-seq data (GATA2, #SRX070876) are aligned to the human reference genome (hg38) and visualized using the Integrative Genomics Viewer (IGV).
(TIFF)

### Acknowledgments

We appreciate the Histopathology Core Facility for supporting the histological experiments.

### Author Contributions

**Conceptualization:** Yuko Shirota, Takashi Moriguchi.

**Data curation:** Yuko Shirota, Shin'ya Ohmori, Takashi Moriguchi.

**Formal analysis:** Yuko Shirota, Shin'ya Ohmori, Takashi Moriguchi.

**Funding acquisition:** Takashi Moriguchi.

**Investigation:** Yuko Shirota, Shin'ya Ohmori, Takashi Moriguchi.

**Methodology:** Yuko Shirota, Shin'ya Ohmori, Takashi Moriguchi.

**Project administration:** Yuko Shirota, Takashi Moriguchi.

**Resources:** Yuko Shirota, Takashi Moriguchi.

**Software:** Takashi Moriguchi.

**Supervision:** James Douglas Engel, Takashi Moriguchi.

**Validation:** James Douglas Engel, Takashi Moriguchi.

**Visualization:** James Douglas Engel, Takashi Moriguchi.

**Writing – original draft:** Takashi Moriguchi.

**Writing – review & editing:** James Douglas Engel, Takashi Moriguchi.

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
