## [Decision Letter · Decision Letter 0]

4 Oct 2024

PONE-D-24-36911GATA2 participates in protection against hypoxia-induced pulmonary hypertensionPLOS ONE

Dear Dr. Moriguchi,

Thank you for submitting your manuscript to PLOS ONE. After careful consideration, we feel that it has merit but does not fully meet PLOS ONE’s publication criteria as it currently stands. Therefore, we invite you to submit a revised version of the manuscript that addresses the points raised during the review process. The study design is not optimal. There are lack of key data to support the conclusion and proper controls. You do not need to carry out the cell-specific KO studies if the total KO exhibit solid phenotype. All other comments must be addressed properly with new experiments and new data. Otherwise, the revision will not be considered for publication. Please submit your revised manuscript by Nov 18 2024 11:59PM. If you will need more time than this to complete your revisions, please reply to this message or contact the journal office at plosone@plos.org. Please include the following items when submitting your revised manuscript:A rebuttal letter that responds to each point raised by the academic editor and reviewer(s). You should upload this letter as a separate file labeled 'Response to Reviewers'.A marked-up copy of your manuscript that highlights changes made to the original version. You should upload this as a separate file labeled 'Revised Manuscript with Track Changes'.An unmarked version of your revised paper without tracked changes. You should upload this as a separate file labeled 'Manuscript'.

We look forward to receiving your revised manuscript.

Kind regards,

You-Yang Zhao

Academic Editor

PLOS ONE

“This work was supported by Grants-in-Aid for Scientific Research (22K06913 to TM) from the Ministry of Education, Culture, Sports, Science and Technology (MEXT) of Japan.”

“We appreciate the Histopathology Core Facility for supporting the histological experiments. This work was supported by Grants-in-Aid for Scientific Research (22K06913 to TM) from the Ministry of Education, Culture, Sports, Science and Technology (MEXT) of Japan.”

“This work was supported by Grants-in-Aid for Scientific Research (22K06913 to TM) from the Ministry of Education, Culture, Sports, Science and Technology (MEXT) of Japan.”

5. We note that Figure 1B in your submission contain [map/satellite] images which may be copyrighted. All PLOS content is published under the Creative Commons Attribution License (CC BY 4.0), which means that the manuscript, images, and Supporting Information files will be freely available online, and any third party is permitted to access, download, copy, distribute, and use these materials in any way, even commercially, with proper attribution. For these reasons, we cannot publish previously copyrighted maps or satellite images created using proprietary data, such as Google software (Google Maps, Street View, and Earth). For more information, see our copyright guidelines: http://journals.plos.org/plosone/s/licenses-and-copyright.

1. You may seek permission from the original copyright holder of Figure 1B to publish the content specifically under the CC BY 4.0 license. 

Reviewers' comments:

Reviewer's Responses to Questions

**Comments to the Author**

1. Is the manuscript technically sound, and do the data support the conclusions?

Reviewer #1: No

Reviewer #2: Yes

2. Has the statistical analysis been performed appropriately and rigorously? 

Reviewer #1: Yes

Reviewer #2: No

3. Have the authors made all data underlying the findings in their manuscript fully available?

Reviewer #1: No

Reviewer #2: Yes

4. Is the manuscript presented in an intelligible fashion and written in standard English?

Reviewer #1: Yes

Reviewer #2: Yes

5. Review Comments to the Author

Reviewer #1: The current reserch describes the role of the transcription factor GATA2 in in the context of pulmonary hypertension and hypoxia-induced vascular damage. The authors found that GATA2 deficiency predisposes the pulmonary vasculature to damage and fibrosis in response to hypoxia, contributing to pulmonary hypertension. The study topic is of interest but the experiments are poorly designed and lack some key data to support the role of GATA2 in PH model.

1.The authors used Rosa26-CreERT to induce systematic GATA2-KO in mice model. It's better to also include cell specific GATA-KO data (such as using Cdh5-CreERT2, Myh11-CreERT2...) to study which cell type contribute to the GATA2 regulation of PH. Moreover, data showing diminished GATA2 expression in the KO mice tissue are needed to demostrate the knockout efficiency.

2. The study don't have the Right ventricular systolic pressure (RVSP) data to support the conclusion of the role of GATA2 in PH. The RVH measuement is flawed since they used formaldehyde-fixed dry hearts in stead of fresh tissues. The narrow histological difference shown in Figure 3 may come from gender or age difference of animals since the authors used 10 to 16-week-old Gata2 CKO and control mice in the hypoxia study.

3. The whole mice hypoxia study need normoxia controls to show the effects of hypoxia. The authors mentioned the housing of normoxic mice in the methods section but no data shown in the manuscript.

4. In figure 2, the authors show histological data of Gata2 GFP knock-in mice. However, there is no information about this strain provided in the Methods section. Do this mice strain show different PH phenotypes under hypoxia?

Reviewer #2: In this article, the authors used animal models, tissue immunohistology analyses, cell culture-based analyses, and bioinformatic analyses with public-available data bases to show the important roles that GATA2 may be involved in pulmonary arterial endothelial cells fate and pulmonary vascular remodeling. This research is a pioneer work in further understanding the progression of pulmonary vascular remodeling in pulmonary hypertension. Although the authors have provided some solid evidence to indicate the roles and functions of GATA2, there are major and minor issues that the authors should address. The authors can also refer to the attached file for revision suggestions.

Major revisions:

1. In Figure 2 and “GATA2 is abundantly expressed in endothelial cells of the pulmonary vasculature” section, the authors used “Gata2 GFP knock-in mice”. However, the development of the strain is not included in the “Methods” section, and it is recommended to include genotyping results in the supplemental figures. Please also do not forget to include the sequences that are used in the genotyping in Table 1.

2. The authors used the word “control” for many groups (Groups 2, 3 and 4) of mice, but the text did not clearly define which control was used for analysis. It is suggested that the authors just use Group name, instead of using “control”, when presenting the data. For example, what control is used in Line 339

3. In addition to Major revision #2 above, the authors claimed that mice were also kept in normoxic condition; however, these mice were not presented in data. These mice are essential controls to verify the success of hypoxia-induced vascular remodeling. The authors should show that G2f/f mice not bearing the CreERT allele and not treated with 4-OHT developed mild pulmonary vascular remodeling after hypoxia exposure, compared to these mice kept in normoxic condition. This data can be included in supplemental data.

4. In addition to Major revision #2 and #3 above, the authors claimed GATA2 is involved hypoxia-induced pulmonary hypertension; however, what the authors have performed only showed the importance of GATA2 in hypoxia-induced pulmonary vascular remodeling. In order to extend the argument to pulmonary hypertension, right ventricular pressure is required, which will be further supported with right ventricular hypertrophy analysis. The authors should either include the right ventricular pressure data, or correctly conclude their results to be limited to hypoxia-induced pulmonary vascular remodeling, but not pulmonary hypertension.

5. The authors showed that Bmp4 was decreased in G2-CKO lung endothelial cells, but to claim the trend “mirrored” siGATA2 knockdown in HMVEC is not entirely correct. The authors should notice that siGATA2-1 knockdown in HMVEC did very minimum effect on Bmp4 gene expression in HMVEC.

Minor revisions:

1. In Line 186 and Figure 5, there is no statistical analysis.

2. In Figure 7B

a. The spelling for change is not right

b. Use Log2(fold change) instead. Explain the meaning of fold change in figure legend.

3. From Line 211 to Line 219, the authors called for Figures 7B and 7C. However, it is not very reader-friendly to call 7C earlier than 7B. Please find a way to make the figure calling in order.

4. Is the induced GATA2 KO in mice global? If it is a global GATA2 KO, did the authors find any phenotype represent myelodysplastic syndromes? Have the authors ever thought of using heterozygous deletion to mimic the GATA2 haploinsufficiency?

5. The authors are suggested to use abbreviations with consistency. For example, Gata2 CKO in Line 349 should be called G2-CKO to avoid confusion.

6. In Line 376, the authors used VWT to represent pulmonary artery wall thickness; however, there is no full name for this abbreviation.

7. From Line 378 to 380, the authors explained to use Ed and Id to calculate wall thickness. In reality, pulmonary artery’s Ed and Id are not consistent. How did the authors average the parameters within one artery is not clear. Please specify.

8. In Line 383 and 384, the authors claimed that they used rabbit anti-GFP and rabbit anti-CD31, while Figure 2C,D showed co-staining of GFP and CD31. If the authors used a specific kit to facilitate this co-staining, please include.

9. From Line 399 to 405, the authors described a TUNEL assay for pulmonary smooth muscle cells. However, there is no data showed in the article.

10. Please also include a brief description of statistical analysis before “Data availability”.

11. Some captions were covered up in Figure 1A

12. Figure 2 legend mentioned panels E and F, but they were not included in the figure.

13. In Figure 3 legend, Line 483, the authors mentioned a “table”, but it is not Table 1. Please include this table after revised.

14. In Figure 4 legend, Line 489, the authors explained they used arrowheads, but there are no arrowheads in Figure 4A

6. PLOS authors have the option to publish the peer review history of their article (what does this mean?). If published, this will include your full peer review and any attached files.

Reviewer #1: No

Reviewer #2: **Yes: **Yuanjun Shen

---

## [Author Response · Author response to Decision Letter 0]

30 Oct 2024

Dear Editor,

I received the following email reply from the Human Protein Atlas (HPA) office regarding permission to publish HPA data in our paper on Oct 11th. We have followed their suggested guidelines, and they have approved our submission. I hope this meets the necessary requirements. 

b.

TM

---------------

Dear 森口　尚,

Thank you for your email and for your interest in the Human Protein Atlas.

You may use the images in your publication if you give proper attribution to the Human

Protein Atlas as stated below.

Please adhere to the following:

1. The image must have attribution to the Human Protein Atlas, e.g.: “Courtesy of Human

Protein Atlas"

2. The image must have a link to the corresponding image on the Human Protein Atlas web,

e.g.: "v22.proteinatlas.org/humancell" . Please exchange "www" with "vXX" where "XX" is

the current version number of the database. Version number is available on the first page

and here: http://www.proteinatlas.org/about/releases .

3. Cite this publication: Uhlen et al (2015). Tissue-based map of the human proteome.

Science. DOI: 10.1126/science.1260419

This is an example:

[Description of image]. Image credit: Human Protein Atlas, www.proteinatlas.org, (Uhlén M

et al, 2015). Image available at the following URL: v22.proteinatlas.org/humancell

The full license and citation policy is found here:

http://www.proteinatlas.org/about/licence

Please note that we do not give exclusive rights. 

The easiest way to download tissue atlas IH images is to right-click on the

thumbnail-image of interest within the image browser and select "Copy link address" and

you will get the url to the large version of the image:

in the example above you will then get this direct url to the image:

https://www.proteinatlas.org/images/19232/45447_B_9_2.jpg

The scale bar cannot be downloaded however, and you have to calculate and add that

yourself. The downloaded image is in scale 0.5μm/pixel.

Plots are in SVG format and can be downloaded using a browser plugin like this for Chrome:

https://chrome.google.com/webstore/detail/export-svg-with-style/dkjdcaddoplepioppogpckelch

efhddi

The downloaded file can then be opened in, for instance, Adobe Illustrator. 

We encourage you to send us your published work where data from the Human Protein Atlas

have been used.

Let me know if we may assist you further.

With kind regards,

Åsa

Ref-Mid: BSjBRx2-Dq8lG-AAAAAN00AAD7GAAATZkUKrz9-contact@proteinatlas.org

---

## [Decision Letter · Decision Letter 1]

26 Nov 2024

GATA2 participates in protection against hypoxia-induced pulmonary vascular remodeling

PONE-D-24-36911R1

Dear Dr. Moriguchi,

We’re pleased to inform you that your manuscript has been judged scientifically suitable for publication and will be formally accepted for publication once it meets all outstanding technical requirements.

Kind regards,

You-Yang Zhao

Academic Editor

PLOS ONE

Additional Editor Comments (optional):

Reviewers' comments:

Reviewer's Responses to Questions

**Comments to the Author**

1. If the authors have adequately addressed your comments raised in a previous round of review and you feel that this manuscript is now acceptable for publication, you may indicate that here to bypass the “Comments to the Author” section, enter your conflict of interest statement in the “Confidential to Editor” section, and submit your "Accept" recommendation.

Reviewer #1: All comments have been addressed

Reviewer #2: All comments have been addressed

2. Is the manuscript technically sound, and do the data support the conclusions?

Reviewer #1: Yes

Reviewer #2: Yes

3. Has the statistical analysis been performed appropriately and rigorously? 

Reviewer #1: Yes

Reviewer #2: Yes

4. Have the authors made all data underlying the findings in their manuscript fully available?

Reviewer #1: Yes

Reviewer #2: Yes

5. Is the manuscript presented in an intelligible fashion and written in standard English?

Reviewer #1: Yes

Reviewer #2: Yes

6. Review Comments to the Author

Reviewer #1: The authors have addressed the reviewers' comments and corrected all the minor issues in the manuscript.

Reviewer #2: With the modifications that the authors made, and additional discussions of their figures, I am overall satisfied.

7. PLOS authors have the option to publish the peer review history of their article (what does this mean?). If published, this will include your full peer review and any attached files.

Reviewer #1: No

Reviewer #2: No

---

## [Editor Report · Acceptance letter]

17 Dec 2024

PONE-D-24-36911R1 

PLOS ONE

Dear Dr. Moriguchi, 

I'm pleased to inform you that your manuscript has been deemed suitable for publication in PLOS ONE. Congratulations! Your manuscript is now being handed over to our production team.

Kind regards, 

on behalf of

Dr. You-Yang Zhao 

Academic Editor

PLOS ONE